# Does Co-Supplementation with Beetroot Juice and Other Nutritional Supplements Positively Impact Sports Performance?: A Systematic Review

**DOI:** 10.3390/nu15224838

**Published:** 2023-11-19

**Authors:** Elida Ferrada-Contreras, Romina Bonomini-Gnutzmann, Carlos Jorquera-Aguilera, Norman MacmiIlan Kuthe, Humberto Peña-Jorquera, Fernando Rodríguez-Rodríguez

**Affiliations:** 1Escuela de Nutrición y Dietética, Facultad de Ciencias, Universidad Mayor, Santiago 8580745, Chile; eliferrada.nutrition@gmail.com (E.F.-C.); rominabonomini@gmail.com (R.B.-G.); 2IRyS Group, School of Physical Education, Pontificia Universidad Católica de Valparaíso, Valparaíso 2374631, Chile; normanmacmillan@hotmail.com (N.M.K.); humberto.apj@gmail.com (H.P.-J.)

**Keywords:** nitric oxide, nutritional enhancement, sports nutrition, training, resistance

## Abstract

In the pursuit of enhanced athletic prowess in different disciplines, athletes constantly look for strategies to increase their physical performance, encompassing technical skills and dietary components, which inevitably, in most cases, include the incorporation of sports supplements. In recent years, there has been an increase in the number of athletes using ergogenic aids. In this context, scientific evidence must play a prominent role in either endorsing or warning against the use of these products, ensuring the preservation of health while promoting the theoretically established positive benefits. In this vein, beetroot juice (BJ) stands out as a key supplement as an ergogenic aid to improve sports performance, given its demonstrated influence on both aerobic and anaerobic exercise. However, despite widespread global demand, there remains a lack of understanding regarding the potential synergistic effects of combining BJ with other supplements. Consequently, our study aims to determine whether the combination of BJ with another nutritional supplement can enhance its beneficial effects and, therefore, optimize physical performance in humans. A comprehensive literature search was conducted in two major databases—Web of Science and PubMed—from 1 January 2018 to 29 January 2023, using specific keywords. After the exclusion criteria, six articles were selected for analysis. Therefore, our study shows that the effectiveness of combining BJ with another supplement mainly depends on the duration of the chronic intervention, which is where the greatest benefits have been observed.

## 1. Introduction

The American College of Sports Medicine (ACSM) emphasizes the importance of selecting and consuming nutrients and supplements tailored to the type of exercise performed for optimal athlete performance [1,2]. In competitive sports, improvements ranging from 0.5% to 1.5% are considered critical for sporting performance [3]. The Australian Institute of Sport (AIS) categorizes supplements based on the level of evidence regarding their impacts on athlete performance, with beetroot juice (BJ) being categorized into Group A [4]. BJ contains a high concentration of inorganic nitrate (NO_3_^−^), a compound found in significant quantities in this food item and as a preservative in processed meats [5].

BJ has a NO_3_^−^ concentration of up to 11.4 g/L [6]. After ingestion, it is reduced to nitrite (NO_2_^−^) through the action of nitrate reductase enzymes by anaerobic bacteria in the oral cavity [7]. Subsequently, NO_2_^−^ is instantly converted into nitrous acid in the stomach and then nitrogen oxides (2HNO_2_ ⟺ H_2_O + N_2_O_3_) [8]. Previous studies have demonstrated that NO_2_^−^ enhances the bioavailability of nitric oxide (NO), a potent vasodilator, via the nitrate–nitrite–nitric oxide pathway, resulting in various physiological functions, such as improved skeletal muscle function and increased cardiorespiratory performance [9,10]. NO acts as a vasodilator in the muscle, where oxygen (O_2_) consumption is highest, allowing local blood flow to adapt to the O_2_ demand within skeletal muscle, thus providing a homogeneous distribution. Supporting this observation, prior research [11] has demonstrated that BJ supplementation, in gel form, prevents a decline in maximal voluntary contraction following three sets of handgrip isotonic exercises. Furthermore, it improves the saturation of muscle O_2_ during exercise recovery and reduces blood lactate levels. Nevertheless, when applying the same protocol for acute dose supplementation, the outcomes differed [12].

Additionally, it not only enhances vasodilation but also regulates cellular respiration and neurotransmission. As a consequence, it enhances other actions such as sprint capacity and cognitive performance [13,14,15,16,17]. In addition, it has been shown to improve performance in high-intensity intermittent efforts with short rest periods due to the faster resynthesis of phosphocreatine [18]. Furthermore, phytonutrients such as betalains and phenolic compounds are believed to be responsible for their antioxidant, anti-inflammatory, and analgesic effects [19,20], thus aiding in the recovery of muscle function [21,22]. Regarding betalains, which are considered among the primary plant inherent bioactive components of beets, the supplementation with 50 mg of betalain-rich concentrates, devoid of nitrate, resulted in a notable reduction in the 10 km race time for triathletes. Furthermore, it demonstrated a more significant decrease in the physiological increase in serum creatine kinase levels and fatigue following the supplementation [23]. The latter is considered key to the performance of athletes.

Therefore, NO_3_^−^ is classified as a nutritional supplement that can directly enhance athletic performance, a stance supported by the International Olympic Committee (IOC) [24], as well as recent reviews [25,26]. Although the evidence in humans is less clear, previous studies in animals [27] observed that NO_3_^−^ exerts a better result on type 2 muscle fibers compared to type 1 fibers. This effect could partially explain the benefits of muscle contraction, reducing the duration of the action potential of the motor unit and more rapidly improving conduction speed, allowing greater calcium release to maintain force production while promoting a decrease in fatigue.

In summary, BJ supplementation is associated with improvements in both aerobic and anaerobic exercises, albeit with limited insight into its potential synergistic impact. The evidence suggests that “it cannot be stated definitively whether the combination of BJ with other supplements has a positive or negative effect, but the effects of BJ supplementation may potentially be influenced by interactions with other supplements, such as caffeine” [9]. 

Although recent research has evaluated the benefits of certain ergogenic aids, including BJ, on athlete performance, these analyses have primarily focused on their isolated effects [28,29]. Our study aims to complement previous investigations by examining the potential synergistic effects of combining BJ with other permissible supplements available on the market.

Based on the foregoing, the objective of this study was to determine whether the combination of BJ with another nutritional supplement can enhance its beneficial effects and, therefore, improve physical performance in humans.

## 2. Material and Methods

### 2.1. Search Strategy

This systematic review adheres to the PRISMA guidelines [30]. Subsequently, the systematic review was registered in PROSPERO (CRD42023425965). A comprehensive literature search was conducted in two major databases—Web of Science and PubMed—considering articles published in the last 5 years (from 1 January 2018, to 29 January 2023). Table 1 outlines the search strategy employed in the Web of Science and PubMed databases.

### 2.2. Selection Criteria

Firstly, the PICO strategy (Cochrane Handbook for Systematic Reviews of Interventions) was used to organize the systematic review with the seven required steps in https://training.cochrane.org (accessed on 2 March 2023). The selection criteria were as follows: (i) articles written in the English language; (ii) articles published in the Web of Science and PubMed databases; (iii) studies involving humans; (iv) original articles, randomized clinical trials, and quasi-experimental studies; (v) participants aged 18 to 65 years old; and (vi) articles published between January 2018 and January 2023. The exclusion criteria were as follows: (i) studies that included individuals with pathologies; (ii) studies that included animals; (iii) studies that included individuals under 18 years old and older adults (+65 years old); (iv) studies unrelated to physical activity and/or exercise; and (v) case studies, case reports, editorials, cross-sectional studies, longitudinal studies, systematic reviews, meta-analyses, and narrative reviews. After removing duplicates, eligibility was assessed by reviewing the title and abstract and, subsequently, reading the full text to evaluate eligibility.

### 2.3. Data Extraction and Reliability

The search was conducted by (E.F.-C.). The titles and abstracts of all retrieved articles were read by (E.F.C., R.B.G., and F.R.-R.). A meeting was held to resolve disagreements regarding eligibility. For each included study, the following information was collected: first author, publication year, study type, main objective, number of subjects, gender, age, body weight, height, or body mass index (BMI) and type of exercise, exercise intervention type, supplementation protocols, evaluation methods, main results obtained, and conclusions. The selected articles were categorized based on the type of supplementation (BJ + citrulline malate, BJ + caffeine, BJ + carbohydrates, BJ + amino acids, BJ + potassium nitrate). Finally, the results of the studies that met the selection criteria were analyzed.

### 2.4. Quality Assessment and Level of Evidence

Bias risk was assessed using the critical appraisal tool for systematic reviews from the Joanna Briggs Institute. This tool includes various checklists specific to the study design. In this case, checklists for randomized clinical trials [31] and quasi-experimental studies [32] were used. Responses for each item had three possible categories: “yes” (criterion met), “no” (criterion not met) and “unclear”. The tools included thirteen items for randomized clinical trials and nine items for quasi-experimental studies. According to the above, studies were considered as “low-quality evidence” when ≤49% of the items were met, “medium-quality evidence” when 50–74% of the items were met, and “high-quality evidence” when ≥75% of the items were met. It should be noted that the responses “not applicable” and “unclear” were excluded from the evaluation, as they do not contribute to the quality of evidence. The three reviewers assessed the quality of the studies separately, and a consensus meeting was organized to resolve differences related to the classification of articles based on criteria fulfilment (Table 2).

## 3. Results

Figure 1 shows the flowchart for the search strategy of the selected articles. A total of 700 studies were found in the two databases. After reading the titles and abstracts of each article, 340 studies were excluded as duplicates, and 126 studies were excluded because they were not relevant to the review topic. Subsequently, the eligibility of a total of 234 studies was assessed. After analyzing the exclusion criteria, finally, six studies were included in the analysis. Six studies had a randomized controlled trial design, and four studies had a quasi-experimental design.

Table 3 provides a summary of the studies included in this review. This review covers data from 106 participants, with sample sizes ranging from 9 to 32 subjects. All studies reported the gender of the participants. None of the studies only included females; only one study included both males and females [34] and five studies only included males [35,36,37,38,39], and All studies reported the age of the participants [34,35,36,37]. The ages of the subjects ranged from 18 to 64 years. The samples were drawn from five different countries: two studies were conducted in Spain, one in Scotland, one in France, one in Iran, and one in Sweden.

Regarding the sample characteristics, four of the six studies focused on athletes such as soccer players [39], triathletes [38], and runners [34,36], while the remaining two studies included physically active individuals in their sample [35,37].

To analyze the effect of supplementation, four studies [36,37,38,39] applied various aerobic tests. Two studies considered strength exercises such as squatting [37] and leg extension [35].

The results revealed that three studies did not show significant effects with the intake of BJ in combination with other supplements like caffeine (CAF) [34,39] and nitrate (N), i.e., N + CIT and N + ARG [35]. In contrast, the other three studies observed positive effects on performance in different tests after the intake of BJ in combination with supplements such as caffeine [37], citrulline [38], and carbohydrates [36].

Additionally, an analysis was conducted to determine the quality of the studies included in this review. For this purpose, the Joanna Briggs Institute criteria checklist was used (Table 2). Different criteria were applied depending on the study characteristics. In this regard, Figure 2 shows that three of the four randomized controlled trials (RCT) met 100% of the criteria [35,37,39], while one met 92.3% [38]. Of the remaining three quasi-experimental studies, two met 77% of the criteria [34,36]. All studies met ≥ 75% of the criteria, classifying them as high quality (HQ).

## 4. Discussion

The objective of this study was to determine whether the combination of beetroot juice (BJ) with another nutritional supplement can enhance its beneficial effects and, therefore, improve physical performance in humans.

Studies of BJ date back to the 1980s and were initially oriented towards clinical health [40,41,42,43,44,45], continuing in this direction until the 1990s [46,47,48,49]. From the year 2000, the first studies of the use of this supplement and its impact on physical exercise began to be published, focusing on its effects on reducing BP and oxygen consumption during exercise [46,47,48,49]. It was primarily determined that its consumption improves tolerance to high-intensity exercise in humans [48]. Subsequently, in 2013, the first systematic reviews on BJ and physical exercise began to appear [50,51].

A variety of studies confirm its utility through different research works [46,47,48,49,50,51], and its intake is even endorsed by the AIS [4] and the IOC [24]. However, it is essential to consider that supplement consumption is increasingly common and often, according to empirical evidence, tends to be combined with others. In this sense, this study aims to determine the effect of BJ intake and its potential enhancement when combined with other ergogenic aids applied to different sports and physical capabilities [37].

### 4.1. Combination of BJ in Aerobic Exercises

A total of four studies included in this review conducted aerobic tests to assess the effects of BJ supplementation in combination with another supplement [34,36,38,39]. Two of these studies combined BJ + CAF in soccer players [39] and runners [34]. In these studies, there were no differences in Yo-Yo test results, VO_2max_, running economy, heart rate, or lactate levels. A prior Australian study conducted in 2014 [52] found differences between the BJ + CAF combination and the placebo but did not find differences between the consumption of CAF alone and the BJ + CAF combination (Mean power 260 + 58 W and 258 + 59 W; *p* = 0.4, respectively). This Australian study suggests that a higher level of the athlete may require chronic BJ intake before a test to benefit performance, and it might work better in sports with higher intensity than just aerobic sports. These two aspects are relevant points to consider in these two studies since they did not achieve effects with the combination [34,39].

On the other hand, the study by Burgos et al. [38] investigated the effects of the combination of BJ + CIT in a group of triathletes. An improvement in aerobic power (*p* = 0.002) was found through chronic consumption of 3 g/day of CIT and 2.1 g/day of NO_3_- over 9 weeks. CIT’s ability to act as a buffer for muscular acidosis has previously been studied, which may help to reduce hyperammonemia and blood lactate accumulation in athletes involved in aerobic or strength activities [53]. In addition to the aforementioned effects of BJ, this would provide a performance boost in high-intensity exercise. Another systematic review [54] reported that CIT supplementation significantly reduced the rating of perceived exertion (7 studies, *p* = 0.03) and muscle soreness 24 h post-exercise (seven studies, *p* = 0.04). However, CIT supplementation did not significantly reduce muscle soreness 72 h post-exercise or lower blood lactate. This is attributed to the acute consumption of CIT (~8 g, 1 h before exercise), as opposed to the chronic administration registered in previous studies that yielded favorable outcomes. Therefore, the results should be considered with caution, as their effects may depend on the specific nature and duration of acute or chronic consumption.

Regarding the combination of BJ + CHO, the study by Burleigh et al. [36] examined the combination of enriched BJ + CHO (Powerade^®^, The Coca-Cola Company, Atlanta, GA, USA) in runners in an incremental test to exhaustion. An improvement in salivary pH was found in the BJ + CHO group compared to the depleted BJ + CHO group, CHO alone, and the control (water) group. This improvement is related to better intestinal substrate absorption efficiency and reduced dehydration, which could enhance physical performance. Most sports drinks have pH levels lower than the acceptable acidity of ~5.5 [55]. Additionally, the negative effect of an acidic and carbohydrate-rich diet could be exacerbated by reduced saliva flow during exercise [56]. In this case, BJ would contribute to preventing saliva acidification caused by the consumption of a sports drink during exercise (Figure 3).

### 4.2. Combination of BJ in Strength Exercises

Regarding the effects of BJ on muscle strength and power, it generally seems to be more effective. In the study by Castillo et al. [37], active men performed countermovement jump (CMJ) tests and squats after the acute consumption of BJ + CAF. The researchers concluded that prior intake of BJ + CAF seems to accelerate energy production capacity after a fatiguing protocol, maintaining CMJ performance in the post-test at 180 s compared to the pre-test. The effect of CAF on performance has been extensively studied. It has been defined that highly trained athletes are more sensitive to CAF and may experience even more improvement [57]. Furthermore, higher doses (6 mg·kg^−1^) may have greater positive effects [37]. However, despite the known distinct benefits of CAF and BJ, the study [37] could not establish a statistical difference between consuming only CAF or only BJ and the BJ + CAF combination. More specific studies are needed to address this research question.

In the study by Burgos et al. [38], which supplemented with 3 g/day of CIT plus 2.1 g/day of BJ (300 mg/day of NIT) for 9 weeks, improved maximal strength and endurance-strength compared to separate CIT or BR supplementation. In this case, the chronic consumption of the BJ + CIT combination had significant positive effects.

On the other hand, the study by Le Roux-Mallouf et al. [35], conducted in 15 healthy men, examined the combination of NIT + CIT and NIT + ARG on isometric knee extension strength. The main results were an increased plasma concentration of NO precursors and enhanced post-ischemic vasodilation, but there was no significant effect on muscle and cerebral oxygenation or peripheral and central neuromuscular fatigue, and no improvement in exercise performance (Figure 3). The use of CIT as a nitric oxide promoter has previously been studied. Trexler et al. [58] studied the acute effects of CIT-Mal on lower limb strength but found its effect to be less pronounced than that of BJ. A recent systematic review and meta-analysis [59] concluded that CIT supplementation alone had a small ergogenic effect during strength training. Another study [60] conducted on trained strength athletes (a single 8 g dose of L-citrulline) did not enhance isometric force production, muscle endurance, or muscle oxygenation parameters. Apparently, acute doses of CIT supplementation are not sufficient to improve muscle oxygenation and strength parameters.

Finally, it is crucial to recognize that the various outcomes observed may have multiple causes, extending beyond the inherent efficacy of the supplement itself. Factors such as the diversity of the sample, encompassing both trained and untrained individuals, as well as the various disciplines under examination, can significantly influence the results. This context suggests that athletes who prefer using supplements might exhibit a higher likelihood of experiencing a placebo effect [61]. Furthermore, a prior study underscores that a combination of motivation, expectancy, and physiological factors can collectively determine whether the outcome is positive or negative [62]. This concept can clearly and eventually interfere as a confounding factor and reduce the gap between the ergogenic effect of a supplement and the placebo effect, diminishing the final statistical power found.

## 5. Conclusions

Few studies have been conducted regarding the combination of beetroot juice with other supplements in recent years. Our analysis indicates that there is evidence of its effectiveness, particularly when exercise intensity is higher. This applies to both aerobic and muscular strength activities. However, there appear to be greater benefits when the combination of beetroot juice with another supplement is consumed chronically. Studies involving the acute intake of two supplements in combination seem to have a lesser impact. Clearly, further research into the combination of beetroot juice with other supplements is needed to address unresolved questions, ultimately improving the health and athletic performance of individuals.

## Figures and Tables

**Figure 1 nutrients-15-04838-f001:**
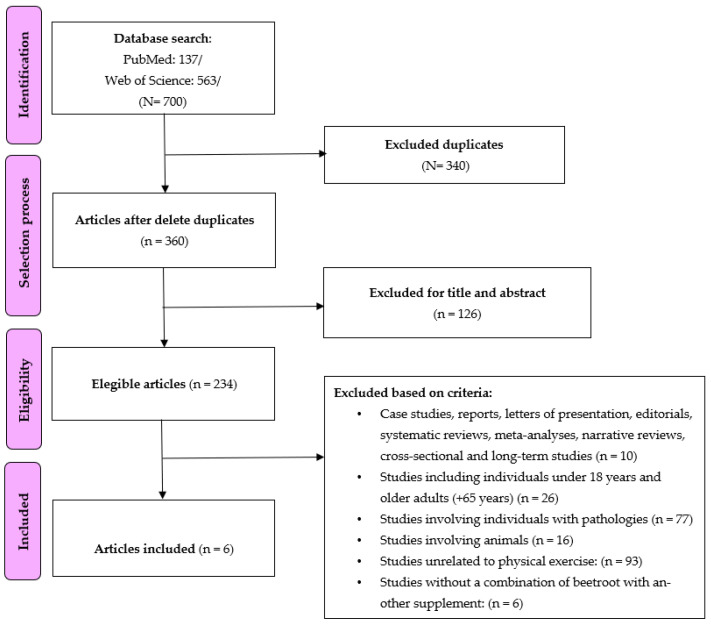
Flow chart used for searching studies.

**Figure 2 nutrients-15-04838-f002:**
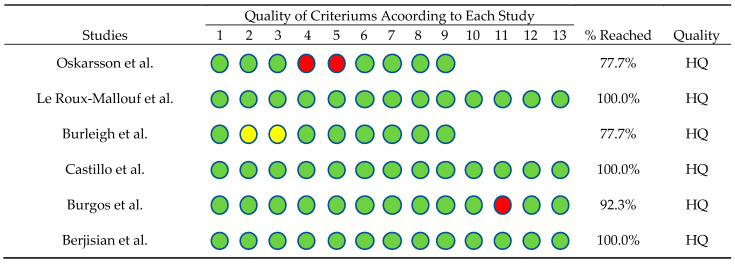
Checklist from the Joanna Briggs Institute’s criteria according to the kind of study, percentage of criteria reached, and quality level of evidence [34,35,36,37,38,39]. Green circles: criterion met, red circles: criterion not met and yellow circles: unclear information.

**Figure 3 nutrients-15-04838-f003:**
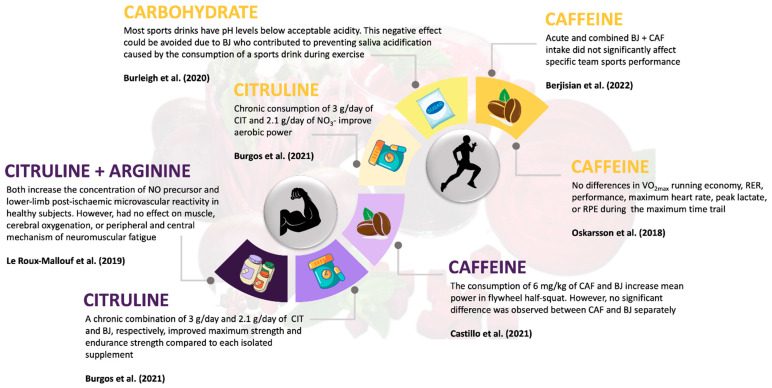
Interplay of BJ and its impact on strength and endurance performance in conjunction with other ergogenic aids [34,35,36,37,38,39].

**Table 1 nutrients-15-04838-t001:** Search strategy in databases.

Databases	Search Strategy	Limits	Filters
Web of Science	ALL = (beetroot juice and supplementation); ALL = (beetroot juice and performance); ALL = (beetroot juice and caffeine); ALL = (beetroot juice and coffee); ALL = (beetroot juice and creatine); ALL = (beetroot juice and beta-alanine); ALL = (beetroot juice and citrulline); ALL = (beetroot juice and protein); ALL = (beetroot juice and amino acid); ALL = (beetroot juice and carbohydrates)	Title Article English	563 filtered elements
PubMed	(Beetroot juice) AND (supplementation); (Beetroot juice) AND (performance); (Beetroot juice) AND (caffeine); (Beetroot juice) AND (coffee); (Beetroot juice) AND (creatine); (Beetroot juice) AND (beta-alanine); (Beetroot juice) AND (citrulline); (Beetroot juice) AND (protein); (Beetroot juice) AND (amino acid); (Beetroot juice) AND (carbohydrates);	Title Article English Age	137 filtered elements

**Table 2 nutrients-15-04838-t002:** Joanna Briggs Institute’s criteria list [33].

RCT Criteria	Quasi-Experimental Criteria
1. Randomization assignment	1. There was clear ‘cause’ and ‘effect’
2. Allocation to treatment groups concealed	2. Participants included in any comparisons were similar
3. Treatment groups similar at the baseline	3. Participants receiving similar treatment/care
4. Participants blind to treatment assignment	4. There was a control group
5. Delivering treatment blind to treatment assignment	5. Multiple measurements taken pre- and post-intervention
6. Outcome assessors blind to treatment assignment	6. Groups adequately described and analyzed
7. Treatment groups treated identically	7. Outcomes measured in the same way for treatment
8. Groups adequately described and analyzed	8. Outcomes measured in a reliable way
9. Participants analyzed in the groups randomized	
10. Outcomes measured in the same way for treatment	
11. Outcomes measured in a reliable way	
12. Appropriate statistical analysis used	
13. Appropriate trial design	

**Table 3 nutrients-15-04838-t003:** Characteristics of the studies analyzed and synthesized.

Author, Year	Study	Objective	Sample	Study Design	Dose	Outcomes
Oskarsson et al., (2018) [34]	Quasi-experimental	Investigate the additive effects of the combination of BJ + CAF during submaximal and maximal short-duration treadmill runs.	*n* = 9 healthy endurance runners 7 men—Age: 30.4 ± 6.3 years Body mass: 73.2 ± 8.3 kg; 2 women—Age: 31.5 ± 9.2 years Body mass: 64.2 ± 1.5 kg	A preliminary test was conducted, followed by four experimental test sessions, which consisted of two submaximal running series of 5 min (at 70% and 80% of VO_2 max_) and a 1 km time trial (TT) in the laboratory. Participants ingested 70 mL of concentrated BJ or without NO_3_^−^, 2.5 h before each test session, and CAF or a PL 45 min before each test session.	BJ group: 7.3 mmol NO_3_^−^ BJ + CAF group: 7.3 mmol NO_3_^−^ + 4.8 ± 0.4 mg/kg CAF	↔ in VO_2max_ in running economy, rest energy rate, heart rate, or rate of perceived exertion (RPE) at the two submaximal intensities (*p* > 0.05). ↔ in performance, maximum heart rate, peak lactate, or RPE during the maximum TT (*p* > 0.05).
Le Roux-Mallouf et al., (2019) [35]	Double-blind, randomized, crossover study.	Compare the effect of different NO precursors on muscle and cerebral oxygenation and peripheral and central neuromuscular fatigue during an isolated knee extension exercise.	*n* = 15 healthy, active men Age: 28 ± 6 years old Body mass: 73 ± 6 kg Height: 179 ± 7 cm	It was evaluated on four occasions after the ingestion of a drink with NO_3_^−^ precursors NO_3_^−^ only, NO_3_^−^ + ARG (arginine), and NO_3_ + CIT or a PL. Separated by ≥4 days per washout period. The following measurements were taken: blood pressure (BP) before and after ingestion; venous blood 65 min after ingestion, followed by an ischemia–reperfusion test in the lower limb to assess NO_3_^−^ dependent vasodilation; near-maximum levels of CIT, ARG, and nitrate-nitrite in blood with isometric knee extension test; and motor cortex excitability, neuromuscular transmission, and muscle contractility.	NO_3_^−^ group: 520 mg NO_3_^−^ NO_3_^−^ + CIT group: 520 mg NO_3_^−^ + 6 g CIT NO_3_^−^ + ARG group: 520 mg NO_3_^−^ + 6 g ARG PL group: 120 mL NO_3_^−^ free	↑ Nitrate [ ] with NO_3_^−^ only, NO_3_^−^ + ARG, and + CIT vs. PL (*p* < 0.001). ↑ ischemia–reperfusion test with NO_3_^−^ + CIT vs. PL. ↔ BP with any supplementation. ↔ in pre-frontal cortex and quadriceps oxygenation, neuromuscular fatigue, or knee extension exercise performance.
Burleigh et al., (2020) [36]	Quasi-experimental	Assess the effects of a dose of BJ (rich in NO_3_^−^) on salivary pH after carbohydrate supplementation at rest and after aerobic exercise.	*n* = 11 trained male runners Age: 30 ± 7 years Body mass: 86.9 ± 14.1 kg Height: 179 ± 7 cm	Participants ingested, 1 h before the test, (a) 140 mL of water (negative control), (b) 140 mL water + CHO (positive control), (c) 140 mL of NO_3_^−^ +CHO, from BJ, or (d) 140 mL depleted BJ NO_3_^−^ + CHO (placebo). During the tests, they ingested 750 mL of water or CHO drink and gel before, during, and after 90 min of submax running. After a 20-minute rest period, they performed 20 min of running at a speed equivalent to 90% of the GET.	Negative control group: 140 mL water Positive control group: 140 mL water + 30 g CHO BJ + CHO group: ~12.4 mmol NO_3_- + 30.8 g CHO PL group: 140 mL depleted BJ NO_3_^−^ + 30.8 g CHO	↑ salivary pH for several hours after ingestion with enriched BJ with NO_3_^−^ + CHO. ↓ saliva acidification that followed the consumption of carbohydrate-rich supplements before and after a sustained period of exercise. For athletes who regularly consume carbohydrates, NO_3_^−^ intake can provide a benefit.
Castillo et al., (2021) [37]	Randomized controlled trial (double-blind, crossover).	To analyze the effects of different supplementation conditions with beetroot juice and caffeine on fatigue and performance in flywheel half-squat tests in men.	*n* = 16 active men. Age: 22.8 ± 4.9 years old Body mass: 74.4 ± 9.6 kg BMI: 23.7 ± 2.4 kg/m^2^	Placebo (PL), caffeine (CAF), beetroot juice (BJ), and BJ + CAF combined were used. To assess the effect of supplementation, participants completed a countermovement jump (CMJ) before (Pre), 30 s after (post-30 s), and 180 s after (post-180 s) completing flywheel half-squats.	PL group: 140 mL (ECO Saludviva) CAF group: 6 mg/kg^−1^ CAF BJ + CAF group: 140 mL BJ + 6 mg/kg^−1^ CAF	↑ mean power (~1000 W, *p* < 0.001) in flywheel half-squats with CAF, BJ, and BJ + CAF vs. PL ↔ on exercise-related fatigue. ↑ recovery with CAF + BJ after a demanding exercise protocol.
Burgos et al., (2021) [38]	Randomized controlled trial (double-blind, placebo-controlled).	To determine the effects of 9 weeks of CIT (citrulline) + BJ supplementation on maximum performance, endurance strength, and aerobic power.	*n* = 32 amateur male triathletes Age: 32.17 ± 4.87 years Body mass: 73.5 ± 5.4 kg BMI: 22.57 ± 1.79 kg/m^2^ Height: 179 ± 8 cm	Six sessions were completed per week for 9 weeks (70% aerobic, 20% strength, and an additional 10% Core), totaling 2.5 h per session. Measurements included horizontal jump (HJUMP), hand dynamometry, 1-min abdominal test, and Cooper’s test. Supplementation included PL or 6 capsules/day	PL group: 3 g/day cellulose BJ + CIT group: 300 mg/day NO_3_^−^ + 3 g/day CIT	↑ maximum strength and endurance strength with CIT + BJ for 9 weeks. ↑ performance (in tests involving aerobic power) compared to supplementation with CIT or BJ alone.
Berjisian et al., (2022) [39]	Randomized controlled trial (double-blind, placebo-controlled).	To assess the acute effects of combined BJ + CAF intake on specific team sports performance, compared to PL, BJ, and CAF.	*n* = 16 semi-professionals footballers’ men Age: 19.8 ± 2.2 years old Body mass: 69.2 ± 6.1 kg Height: 177.3 ± 6.0 cm	The participants ingested a 60 mL bottle of liquid containing NO_3_^−^, L-arginine, and L-ornithine or dry powder without NO_3_^−^, as PL, from the same bottle 2.5 h before the start of the tests. One hour before the test, they consumed one capsule of CAF. This resulted in four experimental trials, which consisted of BJ + CAF, CAF + PL, BJ + PL, and PL + PL.	BJ + CAF group: 6.3 mmol NO_3_^−^ + 5 mg/kg CAF CAF + PL group: 5 mg/kg CAF + 0.015 g NO_3_^−^ BJ + PL group: 6.3 mmol NO_3_^−^ + 0.015 g NO_3_^−^ PL + PL group: 0.015 g NO_3_^−^ + 0.015 g NO_3_^−^	↔ in the Yo-Yo Intermittent Recovery Test Level 1 (YYIR1) among BJ + CAF, CAF + PL, BJ + PL, and PL + PL (*p* = 0.55).

**Abbreviations:** ARG = arginine; BJ = beetroot juice; BP = blood pressure; CAF = caffeine; CIT = citrulline; CMJ = countermovement Jump; HJUMP = horizontal jump test; PL = placebo; RPE = rate of perceived exertion; TT = time Trial; YYIR1 = Yo-Yo Intermittent Recovery Test Level 1. ↑: increase ↓: decrease, ↔: no change.

## Data Availability

Data is available at https://drive.google.com/drive/folders/1ZwdqKtx5V8llIvJOLmZqqn3MO1rdYGWW?usp=share_link.

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
