# Peer review of "Does Co-Supplementation with Beetroot Juice and Other Nutritional Supplements Positively Impact Sports Performance?: A Systematic Review"

_nutrients, 2023, doi:10.3390/nu15224838_

Round 1

Reviewer 1 Report

Comments and Suggestions for Authors

In the manuscript ‘Effects of Beetroot Juice Intake in Combination with Other Nutritional Supplements on Physical Performance: A Systematic Review’ authors determine whether the combination of BJ with another nutritional supplement can enhance its beneficial effects and thereby, optimize physical performance in humans. The authors have done a great job, carrying out a systematic review following the PRISMA principles. The article is well explained and the conclusions are clear.

Comments:

-          The English throughout the article must be widely checked. Countless grammatical and typographical errors (commas and spaces). References must use the MDPI style.

-          The table should be read more comfortably. Articles should be ordered by year of publication. In addition, the authors should indicate the dose received in a column, as well as the NEA at the beginning of the table. For example, BJ+citrulline. It would also include a column of measurements made in each article. For example, countermovement jump for the first paper. Next, I would include a column of main outcomes, using up arrows, down arrows, or the equals symbol to see the results of the articles at a glance.

Author Response

REVIEWER 1

In the manuscript ‘Effects of Beetroot Juice Intake in Combination with Other Nutritional Supplements on Physical Performance: A Systematic Review’ authors determine whether the combination of BJ with another nutritional supplement can enhance its beneficial effects and thereby, optimize physical performance in humans. The authors have done a great job, carrying out a systematic review following the PRISMA principles. The article is well explained and the conclusions are clear.

Comment 1. The English throughout the article must be widely checked. Countless grammatical and typographical errors (commas and spaces). References must use the MDPI style.

Response 1. Thank you for your comment. The language has been checked. The references were adjusted to Nutrients format.

Comment 2. The table should be read more comfortably. Articles should be ordered by year of publication. In addition, the authors should indicate the dose received in a column, as well as the NEA at the beginning of the table. For example, BJ+citrulline. It would also include a column of measurements made in each article. For example, countermovement jump for the first paper. Next, I would include a column of main outcomes, using up arrows, down arrows, or the equals symbol to see the results of the articles at a glance.

Response 2. Thank you for your comments. The articles were ordered according to year of publication.

Table 1 has been modified and some things according to our understanding were added. For example, symbols and a column with dose. Other information was included previously in the table. Thank you for the suggestion, the table has improved.

Reviewer 2 Report

Comments and Suggestions for Authors

In their manuscript “Effects of beetroot juice intake in combination with other nutritional supplements on physical performance: A systematic review”, Ferrada-Contreras and co. address the important and still somehow controversial question about the efficiency of nitrate from beet juice to affect physiological processes in body. There is a substantial body of evidence for improvements, but many studies also show no effect at all. Fortunately, so far there aren’t studies documenting detrimental effect of beet juice on health. The review is well written and summarizes well the current knowledge about the topic. As the authors pointed out, the effect of additional supplementation is mainly seen with chronic beet juice supplementation, which is important fact to keep in mind when considering such a supplementation, especially for therapeutical reasons.

Comments:

1. Introduction page 1, line 43/44: “In stomach…. NO2- is instantly converted into nitrous acid and re-enters circulation as NO2-“. This sounds a little bit confusing because it gives impression that, in net balance, nothing happens. Then why to mention it? Part of nitrite in stomach undergoes acidic disproportionation and converts into NO that can diffuse into bloodstream and likely is oxidized into nitrate. Nitrite itself is likely transported/diffuse into bloodstream… I would appreciate a bit more information in manuscript about this topic.

2. I appreciate clear and storytelling Figure 3. However, it would be useful to also know the amount of BJ/nitrate ingested in each study, not only the supplement amounts.

3. I understand this was not a scope of this review, but similar analysis performed for patients ingesting beet juice for therapeutical reasons would be of high importance.

Author Response

REVIEWER 2

In their manuscript “Effects of beetroot juice intake in combination with other nutritional supplements on physical performance: A systematic review”, Ferrada-Contreras and co. address the important and still somehow controversial question about the efficiency of nitrate from beet juice to affect physiological processes in body. There is a substantial body of evidence for improvements, but many studies also show no effect at all. Fortunately, so far there aren’t studies documenting detrimental effect of beet juice on health. The review is well written and summarizes well the current knowledge about the topic. As the authors pointed out, the effect of additional supplementation is mainly seen with chronic beet juice supplementation, which is important fact to keep in mind when considering such a supplementation, especially for therapeutical reasons. 

Comment 1. Introduction page 1, line 43/44: “In stomach…. NO2- is instantly converted into nitrous acid and re-enters circulation as NO2-“. This sounds a little bit confusing because it gives impression that, in net balance, nothing happens. Then why to mention it? Part of nitrite in stomach undergoes acidic disproportionation and converts into NO that can diffuse into bloodstream and likely is oxidized into nitrate. Nitrite itself is likely transported/diffuse into bloodstream… I would appreciate a bit more information in manuscript about this topic.

Response: Thank you for your comment. We reviewed the literature, and the description is correct. However, we have corrected the sentence that

Comment 2. I appreciate clear and storytelling Figure 3. However, it would be useful to also know the amount of BJ/nitrate ingested in each study, not only the supplement amounts. 

Response 2: Thank you very much for your comment.

We have realized this. We have added that information in detail in a "Dose" column in Table 1.

Comment 3. I understand this was not a scope of this review, but similar analysis performed for patients ingesting beet juice for therapeutical reasons would be of high importance.

Response 3. Thank you very much for your comment.

It's a really interesting topic. Unfortunately, it is not our line of research now, but it can be considered as future work perspectives in people with cardiovascular pathologies.